# Rapid Degradation of Carbon Tetrachloride by Microscale Ag/Fe Bimetallic Particles

**DOI:** 10.3390/ijerph18042124

**Published:** 2021-02-22

**Authors:** Xueqiang Zhu, Lai Zhou, Yuncong Li, Baoping Han, Qiyan Feng

**Affiliations:** 1School of Environmental Science and Spatial Informatics, China University of Mining and Technology, Xuzhou 221116, China; zhoulai99@cumt.edu.cn (L.Z.); fqycumt@126.com (Q.F.); 2Department of Soil and Water Sciences, Tropical Research and Education Center, University of Florida, Homestead, FL 33031, USA; yunli@ufl.edu; 3School of Geography & Geomatics and Urban-Rural Planning, Jiangsu Normal University, Xuzhou 221116, China; bphan@cumt.edu.cn

**Keywords:** microscale bimetallic Ag/Fe, carbon tetrachloride, degradation, reaction kinetics

## Abstract

Cost-effective zero valent iron (ZVI)-based bimetallic particles are a novel and promising technology for contaminant removal. The objective of this study was to evaluate the effectiveness of CCl_4_ removal from aqueous solution using microscale Ag/Fe bimetallic particles which were prepared by depositing Ag on millimeter-scale sponge ZVI particles. Kinetics of CCl_4_ degradation, the effect of Ag loading, the Ag/Fe dosage, initial solution pH, and humic acid on degradation efficiency were investigated. Ag deposited on ZVI promoted the CCl_4_ degradation efficiency and rate. The CCl_4_ degradation resulted from the indirect catalytic reduction of absorbed atomic hydrogen and the direct reduction on the ZVI surface. The CCl_4_ degradation by Ag/Fe particles was divided into slow reaction stage and accelerated reaction stage, and both stages were in accordance with the pseudo-first-order reaction kinetics. The degradation rate of CCl_4_ in the accelerated reaction stage was 2.29–5.57-fold faster than that in the slow reaction stage. The maximum degradation efficiency was obtained for 0.2 wt.% Ag loading. The degradation efficiency increased with increasing Ag/Fe dosage. The optimal pH for CCl_4_ degradation by Ag/Fe was about 6. The presence of humic acid had an adverse effect on CCl_4_ removal.

## 1. Introduction

The application of zero valent iron (ZVI) for environmental remediation of soil and groundwater has been widely studied for more than two decades [1]. Nanoscale zero valent iron (nZVI) exhibits better performance in contaminant removal compared with the microscale ZVI, because nZVI has a larger surface area and higher reactivity due to its small size [2]. Therefore, a wide range of contaminants can be effectively removed by nZVI, such as chlorinated solvents, phenols, pesticides, heavy metals, arsenite, nitrate, etc. [1,2,3]. However, limited mobility of the unmodified nZVI, the rapid aggregation of particles, fast corrosion rate in water, high cost of nZVI production, and potential ecotoxicity restrict its application for contaminant removal [3,4].

Less-expensive microscale iron particles (mZVI) have been considered as an alternative for nZVI [5,6,7]. Compared with nZVI, the use of mZVI in field applications has some advantages, such as a longer lifetime, easier and safer handling of dry particles, lower commercial costs, and lower potential toxicity to the ecosystem [2,8,9]. It was reported that the reactivity of some newly designed mZVIs was similar to highly reactive nZVIs, and even up to one order of magnitude higher according to specific surface-area-normalized reaction rate constants under standardized experimental conditions [6], while the mZVI particles had approximately a 10–30-fold lower corrosion rate than nZVI particles [7]. ZVI can be modified to enhance the reactivity for pollutants by the deposition of transition or noble metals on the surface of ZVI to form bimetallic particles which act as a catalyst, such as Pd, Ni, Cu, Ag, etc. [10,11,12]. The mechanisms for bimetallic systems improving the degradation rate can be explained by: (1) promoting electron transfer by forming galvanic couples; (2) generating reactive hydrogen atoms; and (3) slowing the deposition of corrosion products on the surface of ZVI and protecting the ZVI from passivation [10,13,14]. The contaminant degradation capacity of the iron-based bimetallic system is related to the corrosion of ZVI. The iron corrosion rate is enhanced due to the formation of a galvanic couple between the noble metal and Fe [15]. The potential difference between the metals is the driving force for the corrosive reaction. The greater the potential difference, the faster corrosion occurs. The standard electrochemical potentials for Fe^2+^/Fe, Pd^2+^/Pd, Cu^2+^/Cu, Ni^2+^/Ni, and Ag^+^/Ag are −0.44, 0.92, 0.34, −0.25 and 0.80 V, respectively [14]. Theoretically, the potential difference for the Fe/Pd and Ag/Fe pairs are 1.36 V and 1.24 V, respectively. Pd is the most common reductive dehalogenation catalyst used with ZVI for remediation purposes [16,17]. However, the high cost of Pd limits the practical application of Pd/Fe bimetal. Theoretically, Fe and Ag form a galvanic couple with a higher potential, and subsequently improve electron transfer due to the higher standard potential of Ag. Moreover, Ag is substantially cheaper compared to Pd.

In the present work, micro-scale sponge iron-based Ag/Fe bimetal particles were synthesized and used to remove carbon tetrachloride (CCl_4_) in aqueous solution. The reaction kinetics and the effects of main parameters, such as Ag loading, Ag/Fe dosage, initial pH of solution, and humic acid, on dechlorination efficiency were studied.

## 2. Materials and Methods

### 2.1. Reagents

Chemicals (AgNO_3_, HCl, NaOH, CCl_4_, methanol, ethanol, and sodium humic acid) used for this study were analytical reagent grade and obtained from the Sinopharm Chemical Reagent Company (Shanghai, China), and irregularly-shaped sponge iron particles with a size less than 150 µm from Tianjin Zhongcheng iron powder factory (Tianjin, China) were used in the experiment. Deionized water was used throughout the whole experiment process.

### 2.2. Preparation of Bimetallic Particles

The Ag/Fe bimetallic particles with five different Ag loadings were prepared by the displacement plating. Sponge iron particles of 5 g mass were added to 100 mL AgNO_3_ solution with different concentrations (100, 200, 300, 400 and 500 mg/L), reacted on a rotary shaker at 200 rpm for 30 min and followed by vacuum filtration. Then, the particles were washed with a 4:1 ethanol–water solution. Finally, the bimetallic particles were dried in a vacuum for 12 h at 50 °C. Assuming all of the catalytic metal was reductively precipitated onto the sponge iron, the content of the Ag in the bimetallic reductant was calculated to be as 0.2. 0.4, 0.6, 0.8 and 1.0 wt.%.

### 2.3. Batch Reactor Experiments

Four sets of batch experiments were conducted to investigate the reaction kinetics and effects of Ag/Fe dosage, solution pH and humic acid on the dechlorination rates. The batch experiments were conducted in 100 mL serum vials on a rotary shaker at 25 ± 0.2 °C and 200 ± 5 rpm. The experimental conditions are listed in Table 1.

### 2.4. Analysis Procedure

The samples (100 µL) were taken from the serum vials and then placed in 20 mL headspace vials with 9.9 mL deionized water at selected time intervals. The samples were analyzed by an Agilent 6890N Network Gas Chromaogragh (GC) equipped with a G1888 Network Headspace Sampler and a 30 m HP-5 capillary column. The temperature program of the GC was as follows: oven temperature of 30 °C, injection port temperature of 150 °C, and detector temperature of 250 °C. Separation was conducted with an oven temperature program: initial 30 °C held for 1 min and ramped at 1 °C/min to 80 °C and held for 1 min. Ultrapure nitrogen was used as a carrier gas with a flow rate of 2 mL/min (split ratio 10:1).

## 3. Results and Discussion

### 3.1. Characterization of Ag/Fe Particles

The morphology and structure of Ag/Fe was analyzed by a Quanta 250 environmental scanning electron microscope (Field Electron and Ion Co., Hillsboro, OR, USA) equipped with a QUANTAX 400-10 Energy Dispersive Spectrometer (Bruker, Karlsruhe, Germany). The Ag loading of the bimetal particle was 1.0 wt.%. As shown in Figure 1, the micro-sized particles were unevenly dispersed on the surface, forming a heterogeneous layer. Energy dispersive spectroscopy (EDS) results demonstrated that these micron-sized particles were Ag, which proved the formation of bimetallic catalytic reduction materials. The Ag atomic percentage dispersion on the surface of Fe measured by EDS was 4.56%, which also showed the heterogeneity of Ag loading on the surface of ZVI.

X-ray photoelectron spectroscopy (XPS) could determine the elemental composition and chemical oxidation state at the surface of Ag/Fe. The XPS of fresh 0.4 wt.% Ag/Zn was tested by the Thermo Scientific ESCALAB 250Xi (Thermo Fisher, Waltham, MA, USA) with a pass energy of 20 eV and an X-ray spot of 900 μm. The test results were processed with XPSPEAK4.1, and a Shirley function was used to subtract the background. The peaks of O1s and Fe2p were fitted with Gaussian–Lorentzian curves. The binding energy scale was corrected using the C1s signal of 285.19 eV. As presented in Figure 2, the binding energies of Ag3d_5/2_ and Ag3d_3/2_, peaking at 368.13 and 374.14 eV, respectively, suggested that Ag bound to the surface was zero valent [18]. Two peaks at ~711.08 eV and ~725.03 eV corresponded to Fe 2p_3/2_ and Fe 2p_1/2_, indicating the existence of Fe(III). O1s was decomposed into three peaks at 529.9 eV, 531.2 eV, and 532.3 eV, which corresponded to the binding energy in O^2−^, OH^−^, and chemically or physically adsorbed water, respectively [19]. The ratio of OH to O^2−^ on the Ag/Fe surface was 1.03, in agreement with the bulk FeOOH stoichiometry, indicating that the iron on the surface mainly existed as FeOOH [20]. It was suggested that Ag/Fe bimetallic particles were oxidized during preparation and storage.

### 3.2. Kinetics of CCl_4_ Degradation

Figure 3 showed the degradation of CCl_4_ by Ag/Fe particles with different Ag loadings of 0.2 wt.%, 0.4 wt.%, 0.6 wt.%, 0.8 wt.% and 1.0 wt.%. As can be seen from this figure, Ag/Fe particles could effectively degrade CCl_4_ with the degradation efficiency over 98% within 40 min. In our previous study, CCl_4_ removal by microscale sponge ZVI exceeded 98% after 48 h under the condition of 20 g/L ZVI, 20 mg/L CCl_4_, and pH 7, and the *k*_obs_ was 0.1151 h^−1^ [21]. It clearly demonstrated that the Ag addition had a tremendous effect on the dechlorination rate. The SEM-EDS results showed that Ag was unevenly deposited on the surface of ZVI to form a heterogeneous layer, leading to the nonlinear dependence of reactivity on Ag loading [22]. At higher Ag loadings, the removal efficiency decreased due to larger aggregation of Ag, which reduced the Ag surface area. The degradation ratios of CCl_4_ were 47.9%, 70.0%, 68.9%, 58.2% and 50.9% within 15 min with the Ag loadings of 0.2 wt.%, 0.4 wt.%, 0.6 wt.%, 0.8 wt.% and 1.0 wt.%, respectively. When Ag loading was less than 0.4 wt.%, the increase in Ag loading improved CCl_4_ degradation. An initial increase in catalytic metal loading on the bimetallic particles increased the number of catalytic metal “island” (i.e., galvanic cells) and the total cathodic areas on the iron surface, thus promoting the iron oxidation and consequently the rate and extent of dechlorination [23]. When Ag loading was higher than 0.4 wt.%, the dechlorination of CCl_4_ was negatively influenced. It was attributed to the higher loading of Ag absorbing more hydrogen atoms and thereby decreasing the catalytic reactivity [24]. For example, the percent removal of CCl_4_ decreased from 83.6% to 71.9% when the Ag loading increased from 0.4 wt.% to 1.0 wt.% at 20 min. The finding was also supported by the results of pentachlorophenol (PCP) degradation by millimeter s-Fe/Ag bimetal [22] and tetrabromobisphenol A(TBBPA) degradation by Ag/Fe bimetallic nanoparticles [23]. The PCP degradation rate increased with an increase in Ag^0^ loading from 0.5 wt.% to 5 wt.%, and then decreased with further increases in Ag^0^ loading from 5 wt.% to 15 wt.% [24]. Luo et al. reported that the dehalogenation efficiency of TBBPA increased with the Ag loading and a maximal constant reached at the Ag loading of 3 wt.%, and the efficiency decreased at Ag loading greater than 3 wt.% due to inhibition of H_2_ formation when Ag loading was over 3 wt.% [25]. The dechlorination of CCl_4_ was sensitive to Ag loading, and thereby a small amount of Ag loaded on the ZVI surface greatly enhanced the degradation of CCl_4_. Therefore, an appropriate Ag coverage would favor CCl_4_ dechlorination. In this study, the optimal Ag loading was 0.4 wt.%. The Ag/Fe particles with 0.4 wt.% were used in the following experiments.

As shown in Figure 4, the dechlorination of CCl_4_ by Ag/Fe particles was divided into the slow reaction stage (0–10 min) and the accelerated reaction stage (10–40 min). The two stage reactions followed pseudo first-order reaction kinetics, and the corresponding parameters are shown in Table 2. Stage I was a slow reaction stage. During the preparation and storage, the bimetallic particles were oxidized, and passive films were formed on the surface (its main component was FeOOH, and there may have been a small amount of Fe_2_O_3_), which hindered the corrosion of ZVI. It also limited the mass transfer process between CCl_4_ and bimetal at the liquid/solid interface. As a result, the degradation of CCl_4_ was relatively slower. However, after the bimetallic particles came into contact with water, an automatic reduction process occurred, which dissolved the oxide layer on the surface of the particles. The reactions are shown in Equations (1)–(3) [26,27]. The auto-reduction occurred discontinuously on the oxide layer, resulting in a fracture zone on the particle that allowed the fresh Ag/Fe to directly contact the CCl_4_ in the solution [26], which was conducive to the next stage of accelerated reactions.
αFeOOH + e^−^ + 3H^+^ → Fe^2+^ + 2H_2_O(1)
Fe_2_O_3_ + Fe + 6H^+^ → 3Fe^2+^ + 3H_2_O(2)
12Fe_2_O_3_ + 8H^+^ + 8e^−^ → 8Fe_3_O_4_ + 4H_2_O(3)

Stage II was the accelerated reaction stage. Passive film was removed by autoreduction and the fresh Ag/Fe particles surfaces were exposed to the aqueous solution. Accordingly, the corrosion of the ZVI particles was greatly accelerated. The generated hydrogen was adsorbed on the catalytic Ag of the particles surface and formed a more active adsorbed atomic hydrogen (H_ads_) [28]. The CCl_4_ degradation at this stage was mainly achieved by the indirect catalytic reduction of H_ads_ and the direct reduction on the ZVI surface [29], as shown in Equations (4)–(8). The degradation rate of CCl_4_ in the accelerated reaction stage was 2.29–5.57-fold faster than that in the slow reaction stage, and 57.65–66.15-fold faster than that in the sponge ZVI. The change of *k*_obs_ under different loading in stage I was not obvious, while the change of *k*_obs_ under different loading in stage II was significant.

(a)Direct reduction by ZVI:

Fe^0^ + CCl_4_ + H^+^ → Fe^2+^ + CHCl_3_ + Cl(4)

(b)Reduction by catalytic hydrogenation:

Transfer of electron: Fe → Fe^2+^ + 2e^−^(5)

2H_2_O + 2e^−^ → H_2_ + 2OH(6)

Activation: 2M (Fe–Ag) + H_2_ → 2M–H*(7)

Hydrogenation: M–H* + CCl_4_ → M + CHCl_3_ + H^+^(8)

Chloroform (CF) was detected as a degradation intermediate. CF concentration slowly increased with the degradation of CCl_4_. In a 0.4 wt.% Ag/Fe system, the CF concentration reached a maximum after 20 min of reaction, which was 36.5% of the initial molar concentration of CCl_4_; then, the CF concentration gradually decreased. The concentration of CF was only 23.1% of the initial molar concentration of CCl_4_ at 40 min of reaction (Figure 5).

### 3.3. Factors Affecting CCl_4_ Degradation with Ag/Fe Bimetallic Particles

#### 3.3.1. Effect of Ag/Fe Dosage on the CCl_4_ Degradation

Figure 6 showed the effect of Ag/Fe dosage on the degradation of CCl_4_. The CCl_4_ degradation efficiencies were 28.6%, 44.8%, 74.0%, 86.4% and 91.3% for Ag/Fe dosages 5, 10, 20, 30 and 40 g/L within 20 min, respectively. Over 99.0% of CCl_4_ was degraded for all dosages within 40 min. The removal efficiency and rate of CCl_4_ increased with the increase in the dosage. The degradation of CCl_4_ under different Ag/Fe dosages conformed to the pseudo first-order reaction kinetics. The corresponding reaction parameters are listed in Table 3. The degradation of CCl_4_ could also be described by a two-stage first-order kinetic equation, and the *k*_obs_ in stage II were significantly higher. When the Ag/Fe dosage increased from 5 g/L to 40 g/L, *k*_obs_ increased from 0.0767 min^−1^ to 0.1556 min^−1^ in stage II. The degradation of CCl_4_ occurred at the surface of the bimetallic particles; therefore, an increase in the Ag/Fe dosage simultaneously increased the number of active sites, reactive surface area, and the amount of the catalytic Ag, leading to enhanced CCl_4_ degradation [30]. There was a good linear relationship between the dosage and *k*_obs_ at the dosage lower than 30 g/L (Figure 7). It should be noted that when the dosage was increased from 30 g/L to 40 g/L, the increase in *k*_obs_ had a slight decreasing tendency.

To further understand the effect of Ag/Fe dosage on CCl_4_ degradation, a Ag-normalized rate constant was adopted. Similar to the surface area normalized reaction rate constant (*k*_SA_), *k*_Ag_ was defined as *k*_Ag_ = *k*_obs_/Ag dosage, which depicted the degradation efficiency of Ag to CCl_4_. *k*_Ag_ was related to the Ag loading on the ZVI surface and the H_2_ amount in solution. When the initial concentration of targeted pollutant and Ag loading were fixed, *k*_Ag_ was mainly controlled by the effective H_2_ amount produced by ZVI corrosion. Table 4 demonstrated that *k*_SA_ and *k*_Ag_ decreased with the increasing dosage. In stage II, *k*_SA_ decreased from 0.1649 L to 0.0418 L min^−1^ m^−2^ and *k*_Ag_ decreased from 3.8350 to 1.1942 L min^−1^ g^−1^ with an increase in Ag dosage from 0.02 to 0.16 g/L. It showed that CCl_4_ could be totally removed at the dosage of 5 g/L Ag/Fe, but it took a longer reaction time.

Figure 8 shows the changes in CF concentration. CF concentration initially increased with the degradation of CCl_4_, then gradually decreased after the degradation of CCl_4_ was completed.

#### 3.3.2. Effect of pH on CCl_4_ Degradation

Solution pH affected the performance of ZVI (corrosion at low pH, passivation at high pH) and hydrodechlorination reactions on the catalytic metal surface [25]. Tian et al. investigated the effect of pH on the DDT degradation by Ni/Fe nanoparticles. It was found that weaker acidic (4 ≤ pH < 7) and alkaline (7 < pH ≤ 10) reaction conditions were more favorable to the fast degradation of DDT. ZVI disappeared quickly due to strong acidic corrosion at pH < 4, and ferrous hydroxide precipitation occurred on the ZVI surface at pH > 10, both of which were not conducive to the DDT degradation [31]. CCl_4_ degradation by Ag/Fe at 40 min were 90.1%, 94.2%, 99.2%, 97.9%, 73.4% and 26.3% under pH 4, 5, 6, 7, 8 and 9, respectively (Figure 9). The results showed that weakly acidic conditions with pH around 6 were most favorable for the dechlorination of CCl_4_, and lower pH or higher pH were not beneficial for the dechlorination of CCl_4_ by Ag/Fe. At a relative low pH, the presence of H^+^ led to the accelerated corrosion of nZVI, resulting in less precipitation of iron oxide on the surface of iron, thus increasing the removal of CCl_4_ [32]. Lower pH (less than 5) promoted the corrosion of ZVI and accelerated excessive hydrogen generation (Fe^0^ + 2H^+^ → Fe^2+^ + H_2_). Tiny bubbles of H_2_ covered the Ag/Fe surface, and thus inhibited the contact between Ag/Fe and target pollutants and reduced the effective reaction area of Ag/Fe surface [33]. When pH > 7, there was insufficient H^+^ in the system to participate in the dechlorination of CCl_4_, and the electrons released by ZVI were more easily consumed by dissolved oxygen in the water (O_2_ + 2H_2_O + 4e^−^ → 4OH^−^) [23]. Higher pH with numerous OH^−^ ions accelerated the formation of ferrous and ferric oxides and precipitated onto the Ag/Fe surface and occupied the reaction site; thus decreasing the overall reaction rate [34,35].

XPS analysis of Ag/Fe particles after reaction at different pH values of 6.0, 7.0, and 8.0 (Figure 10) showed that the binding energies of Ag3d5/2 and Ag3d3/2 were ~368.24, ~368.20, ~368.22 eV and ~374.23, ~374.18, ~374.20 eV, respectively, indicating that the existence of Ag on the ZVI surface was Ag^0^ after the reaction. There were three chemical species of oxygen on the ZVI surface after the reaction: the binding energies of O1s were ~530.0 eV, ~531.0 eV, and 532.3 eV, which signified that oxygen existed as iron oxides. Oxygens with the binding energies of ~530.0 eV and ~531.0 eV were lattice oxygen in iron oxide and hydroxyl oxygen, respectively [36]. Photoelectron peaks at ~711 eV and ~725 eV corresponded to the binding energies of Fe 2p3/2 and Fe 2p1/2, respectively, suggesting that iron presented as Fe(III) on the particle surface. A strong O1s signal was detected, indicating that major compounds on the Ag/Fe surface were ferric oxides. The peak area ratios of OH^−^ to O^2−^ at pH 6.0, 7.0, and 8.0 were 1.93, 0.89, and 0.55, respectively, indicating that Fe on the particle surface was likely to be in the form of FeOOH and Fe_2_O_3_ at pH 8.0; at pH 7.0, the main compound on the ZVI surface was FeOOH [19,20]. This result revealed that the passive film was more easily formed on the ZVI surface under higher pH, thereby hindering the dechlorination of CCl_4_. In addition, it was worth noting that the Ag on the particle surface was lost after the reaction, and the lower the pH, the more serious the loss. The atomic percentage of Ag decreased from 2.51% (before the reaction) to 0.18%, 0.67% 0.71% (after the reaction) at pH 6.0, 7.0 and 8.0, respectively.

#### 3.3.3. Effect of Humic Acid on CCl_4_ Degradation

Humic substances occur ubiquitously in groundwater [37]. Contaminant removal with ZVI might be influenced by humic substances through enhanced solubilization of contaminants, enhanced sorption, competitive sorption or mediated electron transfer [38]. Figure 11 showed the effect of humic acid (HA) on the degradation of CCl_4_ by the Ag/Fe bimetal. The presence of HA inhibited the removal of CCl_4_ by Ag/Fe, and the inhibition increased with the increase in HA concentration. For the systems with the HA at 0, 5, 10, and 25 mg/L, the *k*_obs_ of CCl_4_ degradation were 0.1220 min^−1^, 0.0585 min^−1^, 0.0472 min^−1^, and 0.0305 min^−1^, respectively. A similar effect of HA on iron-based bimetal reactivity was reported by Tratnyek et al. (2001), Doong et al. (2005) and Yi et al. (2019) [38,39,40]. Tratnyek et al. found that the reduction rate of TCE by ZVI was inhibited by NOM due to the competitive sorption onto the surface of ZVI [38]. Yi et al. (2019) suggested that the adsorption of HA occupied the surface of Ni/Fe nanoparticles and thus hindered the degradation of BDE209 by Ni/Fe [40]. Doong et al. proposed that addition of humic acid decreased the reactivity of palladized irons for PCE due to competition of humic acid for reactive sites on the surface of palladized irons [39].

## 4. Conclusions

The load of Ag in Ag/Fe was preferably 0.4 wt.%. Ag/Fe degradation of CCl4 was divided into a slow reaction stage and accelerated reaction stage, and both phases followed the pseudo-first-order reaction kinetics. The degradation rate of CCl_4_ in the accelerated reaction stage of Ag/Fe system was 2.29~5.57-fold faster than that in the slow reaction stage. For Ag/Fe (0.4 wt.%) system, at pH 6, the degradation rate of CCl_4_ was the highest. Too high or too low pH was not conducive to the degradation of CCl_4_. XPS results indicated that the Ag on the particle surface is lost after the reaction. The degree of loss of Ag increased with decreasing pH; the presence of HA inhibited the reduction and degradation of CCl_4_ by Ag/Fe, and the inhibition increased with the increase in HA concentration.

The iron-based microscale particle with a transition metal developed in this study possesses the ability to substantially accelerate the dechlorination reaction and can be used to degrade organic contaminants such as CCl_4_. Additionally, it has a long life span, low ecotoxicity, and can be prepared economically.

## Figures and Tables

**Figure 1 ijerph-18-02124-f001:**
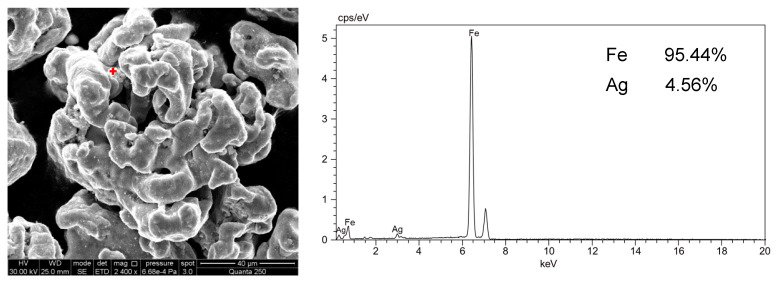
SEM image and EDS of fresh Ag/Fe particle with 1.0 wt.% Ag loading.

**Figure 2 ijerph-18-02124-f002:**
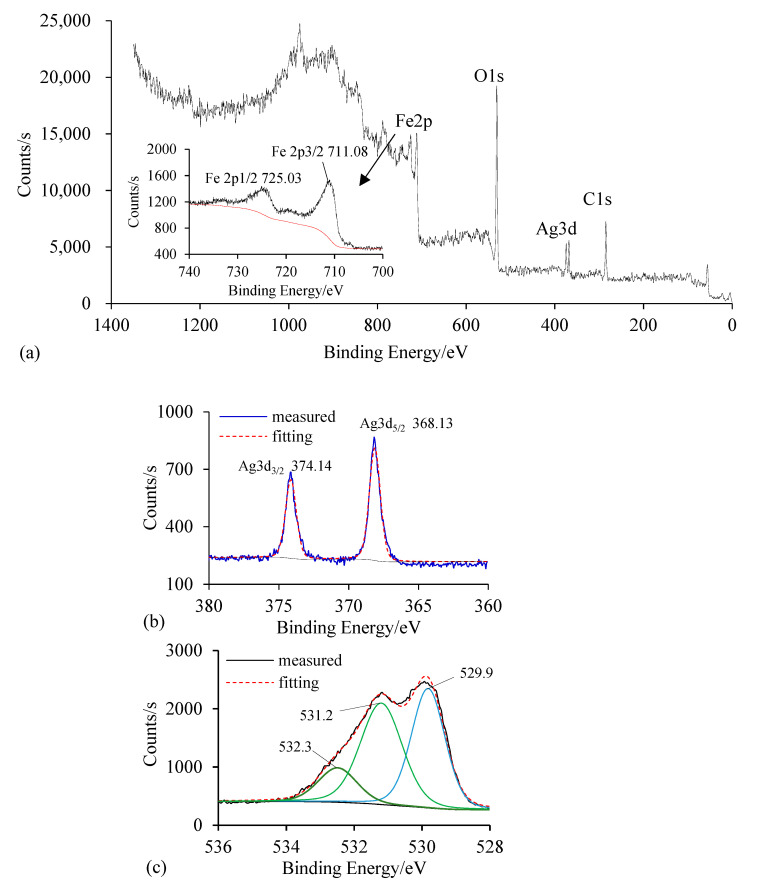
XPS wide-scan and high-resolution spectra of fresh Ag/Fe particle with 0.4 wt.% Ag loading. (**a**) Wide scan spectra of Ag/Fe; (**b**)Ag 3d core level XPS spectra of Ag/Fe; (**c**) O1s core level XPS spectra of Ag/Fe.

**Figure 3 ijerph-18-02124-f003:**
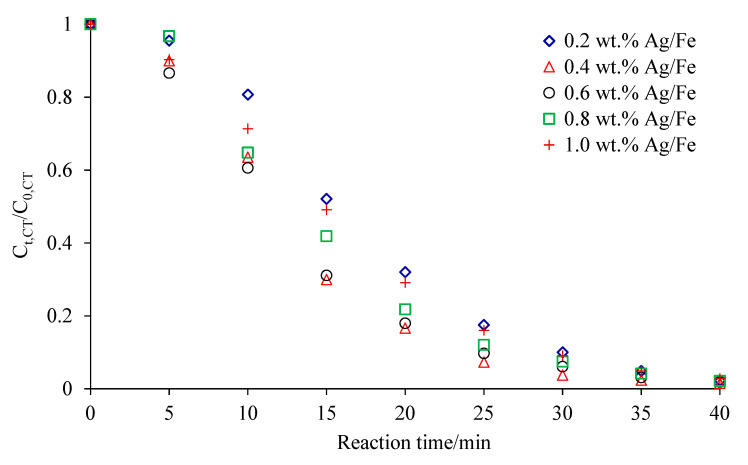
CCl_4_ degradation by bimetallic Ag/Fe with different Ag loadings under the conditions of Ag/Fe dosage of 20 g/L, initial CCl_4_ concentration of 20 mg/L, initial solution pH of 7, and stirring rate of 200 rpm.

**Figure 4 ijerph-18-02124-f004:**
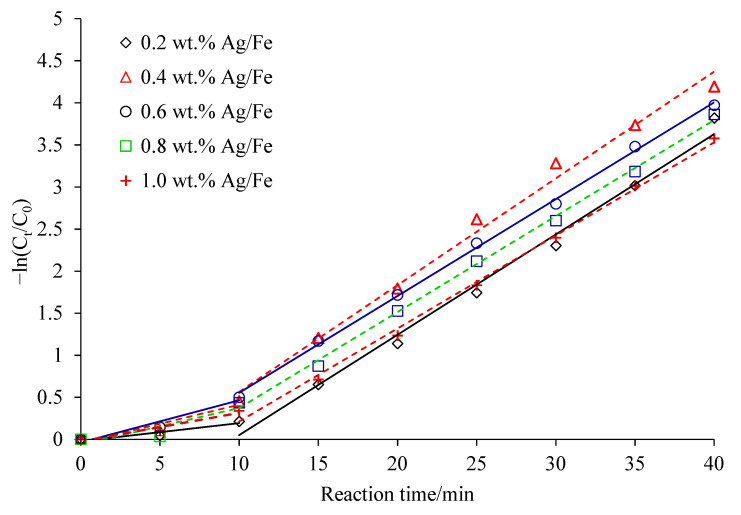
Reaction kinetics for CCl_4_ degradation by Ag/Fe particles. Reaction conditions: Ag/Fe dosage of 20 g/L, initial CCl_4_ concentration of 20 mg/L, initial solution pH of 7, and stirring rate of 200 rpm.

**Figure 5 ijerph-18-02124-f005:**
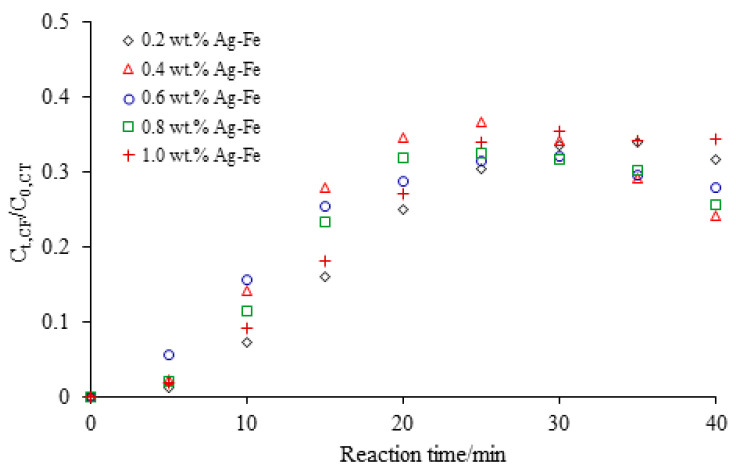
Changes in chloroform concentration at 20 g/L Ag/Fe particles with different Ag loading. C_0,CT_ is the initial CCl_4_ molar concentration, and C_t,CF_ is the CHCl_3_ molar concentration at time t.

**Figure 6 ijerph-18-02124-f006:**
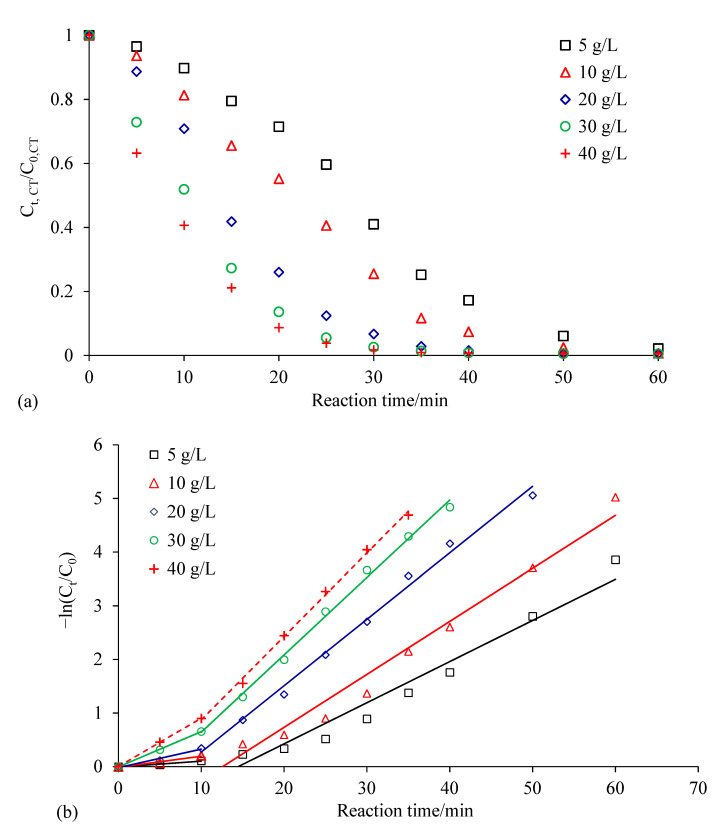
Effect of Ag/Fe dosage on CCl_4_ degradation: (**a**) changes in CCl_4_ concentration with time; (**b**) kinetics of CCl_4_ degradation. Reaction conditions: 0.4 wt.% Ag/Fe dosage of 20 g/L, CCl_4_ initial concentration of 20 mg/L, initial solution pH of 7, stirring rate of 200 rpm. C_0_ is the initial CCl_4_ concentration, and C_t_ is the CCl_4_ concentration at time t.

**Figure 7 ijerph-18-02124-f007:**
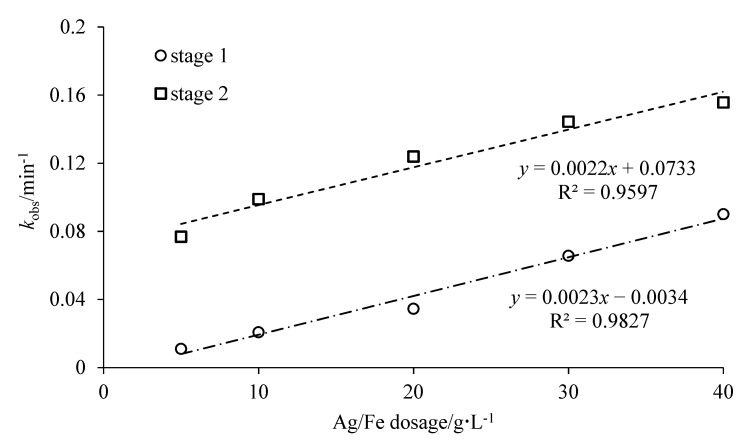
Correlations between the rate constants (*k*_obs_) of CCl_4_ degradation and the Ag/Fe dosage.

**Figure 8 ijerph-18-02124-f008:**
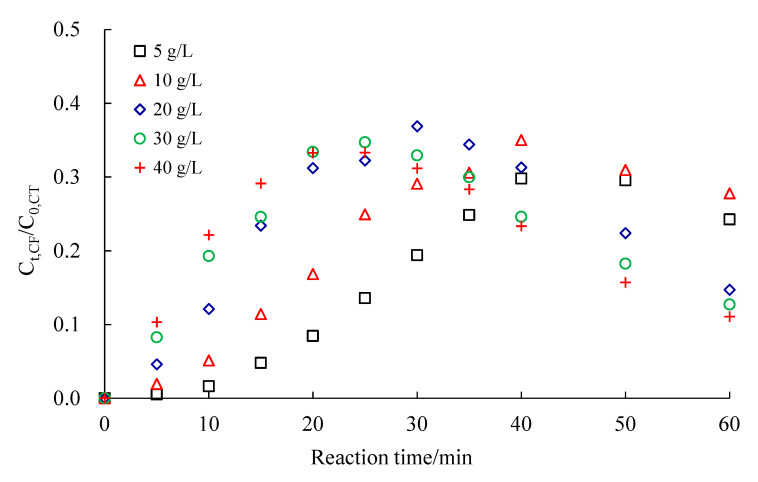
Changes in chloroform concentration with 0.4 wt.% Ag/Fe particles at different dosages. C_0,CT_ is the initial CCl_4_ molar concentration, and C_t,CF_ is the CHCl_3_ molar concentration at time t.

**Figure 9 ijerph-18-02124-f009:**
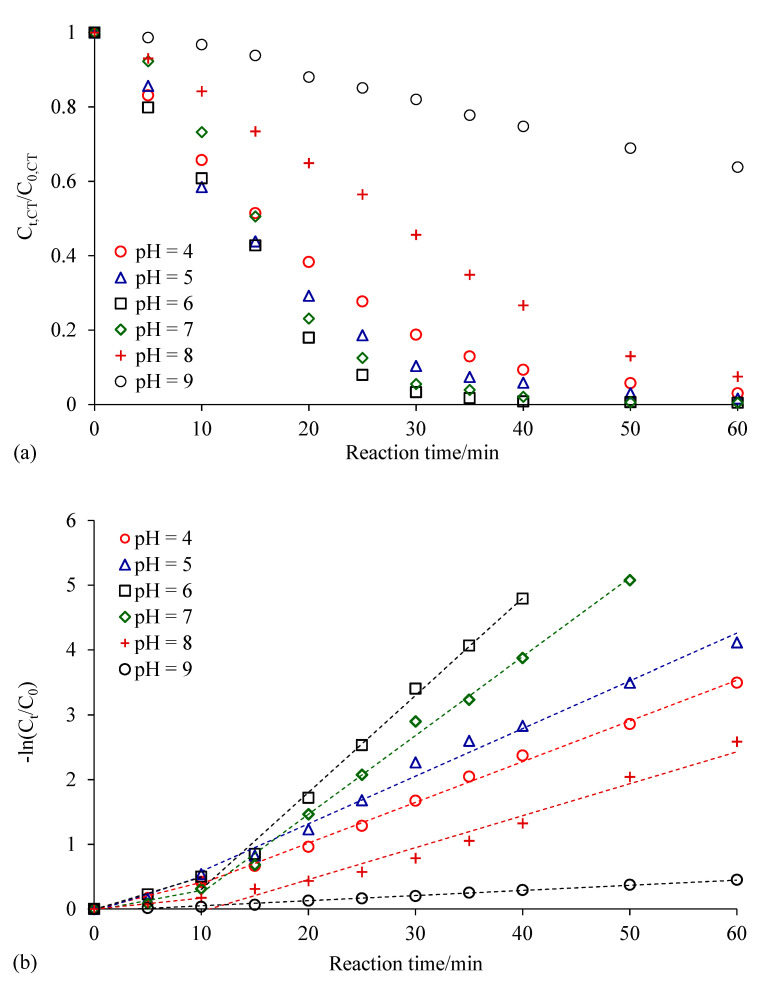
Effect of pH on CCl_4_ degradation by Ag/Fe particles. (**a**) Changes in CCl_4_ concentration with time; (**b**) kinetics of CCl_4_ degradation. Reaction conditions were: initial solution pH of 5, 6, 7 and 8, 0.4 wt.% Ag/Fe dosage of 20 g/L, CCl_4_ initial concentration of 20 mg/L, and stirring rate of 200 rpm. C_0_ is the initial CCl_4_ concentration, and C_t_ is the CCl_4_ concentration at time t.

**Figure 10 ijerph-18-02124-f010:**
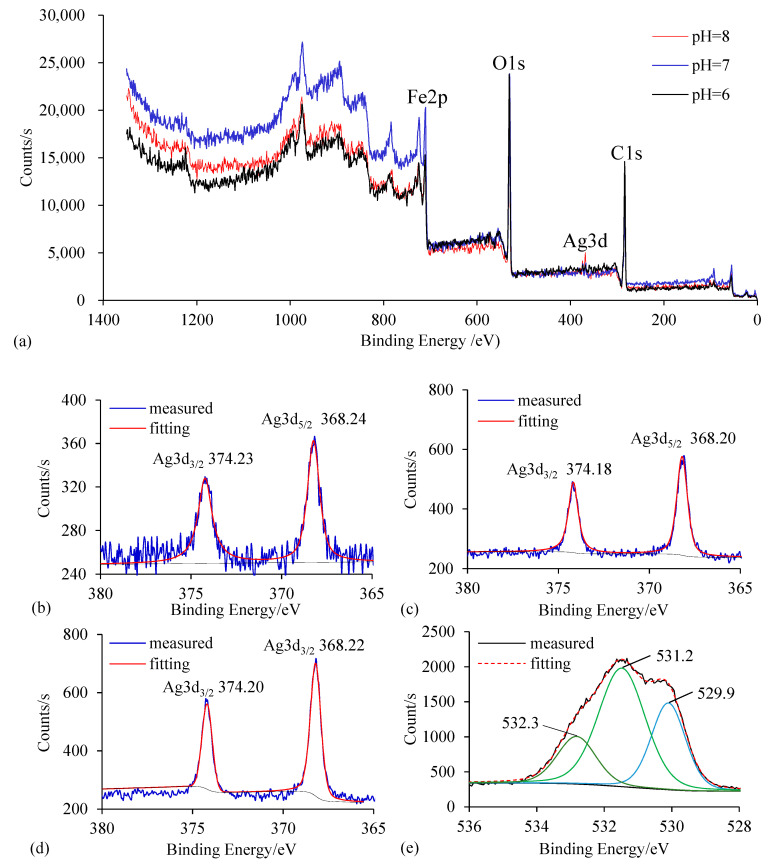
XPS wide-scan and high-resolution spectra of 0.4 wt.% Ag/Fe after reacting with 20 mg/L CCl_4_ solution at initial pH of 6, 7 and 8. (**a**) Wide scan spectra of Ag/Fe; (**b**–**d**) Ag 3d core level XPS spectra of Ag/Fe at pH of 6, 7 and 8 respectively; (**e**–**g**) O1s core level XPS spectra of Ag/Fe at pH of 6, 7 and 8 respectively.

**Figure 11 ijerph-18-02124-f011:**
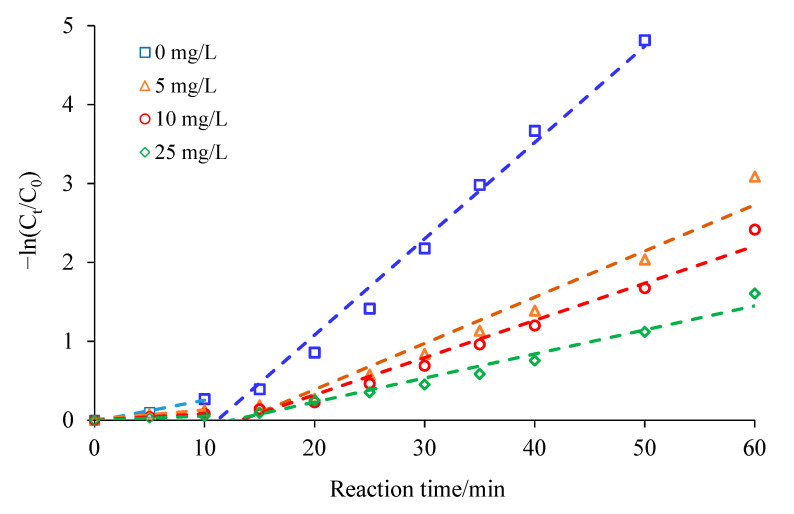
Effect of humic acid on CCl_4_ degradation by Ag/Fe particles. Reaction conditions were: humic acid concentrations of 5, 10 and 25 mg/L, 0.4 wt.% Ag/Fe dosage of 20 g/L, CCl_4_ initial concentration of 20 mg/L, initial solution pH of 7, stirring rate of 200 rpm. C_0_ is the initial CCl_4_ concentration, and C_t_ is the CCl_4_ concentration at time t.

**Table 1 ijerph-18-02124-t001:** Experiment sets for CCl_4_ degradation by Ag/Fe particles.

Experiment Set	pH	CCl_4_ Concentration (µg/L)	Ag/Fe Dosage (g/L)	Ag Loading on Ag/Fe (wt.%)	Humic Acid Concentration (mg/L)
#1	7.0	2.0 × 10^4^	20	0.2, 0.4, 0.6, 0.8, 1.0	0
#2	7.0	2.0 × 10^4^	5, 10, 20, 30, 40	0.4	0
#3	4.0, 5.0, 6.0, 7.0, 8.0	2.0 × 10^4^	20	0.4	0
#4	7.0	2.0 × 10^4^	20	0.4	0, 5, 10, 20

**Table 2 ijerph-18-02124-t002:** Rate constants for CCl_4_ degradation by 20 g/L Ag/Fe particles with different Ag loadings.

Ag Loading		0.2 wt.%	0.4 wt.%	0.6 wt.%	0.8 wt.%	1.0 wt.%
**Stage I**	Equation	−ln(C_t_/C_0_) = 0.0214t − 0.0204	−ln(C_t_/C_0_) = 0.0454t − 0.0404	−ln(C_t_/C_0_) = 0.0501t − 0.0356	−ln(C_t_/C_0_) = 0.0435t − 0.0612	−ln(C_t_/C_0_) = 0.0338t − 0.0224
	*k*_obs_ (min^−1^)	0.0214	0.0454	0.0501	0.0435	0.0338
	*R* ^2^	0.902	0.924	0.943	0.929	0.950
**Stage II**	Equation	−ln(C_t_/C_0_) = 0.1193t − 1.1412	−ln(C_t_/C_0_) = 0.1269t − 0.7044	−ln(C_t_/C_0_) = 0.1151t − 0.5962	−ln(C_t_/C_0_) = 0.1141t − 0.7679	−ln(C_t_/C_0_) = 0.1106t − 0.8928
	*k*_obs_ (min^−1^)	0.1193	0.1269	0.1151	0.1141	0.1106
	*R* ^2^	0.990	0.991	0.998	0.998	0.996

**Table 3 ijerph-18-02124-t003:** Rate constants for CCl_4_ degradation with 0.4 wt.% Ag/Fe at different dosages.

Dosage (g/L)		5	10	20	30	40
**Stage I**	Equation	−ln(C_t_/C_0_) = 0.0109t − 0.0062	−ln(C_t_/C_0_) = 0.0207t − 0.0126	−ln(C_t_/C_0_) = 0.0345t − 0.0174	−ln(C_t_/C_0_) = 0.0656t − 0.0037	−ln(C_t_/C_0_) = 0.0900t − 0.0092
	*k*_obs_ (min^−1^)	0.0109	0.0207	0.0345	0.0656	0.0900
	*R* ^2^	0.962	0.957	0.970	0.999	0.999
**Stage II**	Equation	−ln(C_t_/C_0_) = 0.0767t − 1.1079	−ln(C_t_/C_0_) = 0.0988t − 1.2418	−ln(C_t_/C_0_) = 0.1238t − 0.9669	−ln(C_t_/C_0_) = 0.1443t − 0.8014	−ln(C_t_/C_0_) = 0.1556t − 0.6848
	*k*_obs_ (min^−1^)	0.0767	0.0988	0.1238	0.1443	0.1556
	*R* ^2^	0.9521	0.971	0.993	0.996	0.998

**Table 4 ijerph-18-02124-t004:** Normalized rate constant for CCl_4_ degradation by bimetallic Ag/Fe at different dosages.

Ag/Fe Dosage (g/L)	Ag Dosage (g/L)	*k*_obs_ (min^−1^)		*k*_SA_^1^ (L/(min·m^2^))		*k*_Ag_^2^ (L/(min·m^2^))		*R* ^2^	
		Stage I	Stage II	Stage I	Stage II	Stage I	Stage II	Stage I	Stage II
5	0.02	0.0109	0.0767	0.0234	0.1649	0.5450	3.8350	0.9624	0.9521
10	0.04	0.0207	0.0988	0.0223	0.1062	0.5175	2.4700	0.9575	0.971
20	0.08	0.0345	0.1238	0.0185	0.0666	0.4313	1.5475	0.9702	0.9933
30	0.12	0.0656	0.1433	0.0235	0.0514	0.5467	1.1942	0.9996	0.996
40	0.16	0.0900	0.1556	0.0242	0.0418	0.5625	0.9725	0.9999	0.9978

^1^ The specific surface area of the Ag/Fe particle was 0.093 m^2^/g, measured by the ASAP2460 Surface Area and Porosimetry System, Micromeritics Instrument Corporation, USA. ^2^
*k*_Ag_ = *k*_obs_/Ag.

## Data Availability

The data presented in this study are available on request from the corresponding author.

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
