# Peer review of "Rapid Degradation of Carbon Tetrachloride by Microscale Ag/Fe Bimetallic Particles"

_ijerph, 2021, doi:10.3390/ijerph18042124_

Round 1
Reviewer 1 Report
The article from Zhu et al titled “Rapid Degradation of Carbon Tetrachloride by Microscale Ag/Fe Bimetallic Particles” attempted to fabricate a micro-scale sponge iron based Ag/Fe bimetal which was later employed to remove carbon tetrachloride (CCl4) in an aqueous solution. They went on to demonstrate and systematically analyze the reaction kinetics and the effects of main parameters, such as Ag loading, Ag/Fe dosage, initial pH of solution, and humic acid, on dechlorination efficiency in the same work. This work is systematically done and well presented.
The article is recommended for publication in Int. J. Environ. Res. Public Health after the following minor comments In the abstract, the authors should define abbreviations on first mention. ZVI in the abstract was not defined yet continuously was used in the abstract.
Also, I recommend the authors to give some background statement that motivated this work as the first sentence of the abstract. They did a good work in the introduction section, but in the abstract they just dived into what they did.
In the introduction section, the authors have missed citations on various critical sentences claiming many important things. Just an example see below. “The application of zero valent iron (ZVI) for environmental remediation of soil and groundwater has been widely studied over the last two decades” “Nanoscale zero valent iron (nZVI) exhibits better performance in contaminant removal compared with the microscale ZVI because nZVI has a larger surface area and higher reactivity due to its small size” “Recently, less expensive micro iron particles (mZVI) are considered as an alternative for nZVI.” Citations are needed on these statements.And many others not given here.
Under the Materials and Methods section, some language issues with missing verbs in the statement. Please check well the word “were” just before purchased is missing.
I recommend the authors to proof read again to iron out all these grammar and language issues as they exist throughout the manuscript.
The experimental and discussion sections are well on point and very excellently articulated, good work from the authors.
Author Response
Thank you very much for your comments and suggestions. We have revised our manuscript marked with red after reading your comment.
Comment 1: The authors should define abbreviations on first mention. ZVI in the abstract was not defined yet continuously was used in the abstract.
Response 1: We defined the ZVI in the first sentence of the abstract. “The cost-effective zero valent iron (ZVI) based bimetallic particle is a novel and promising technology for contaminant removal.”
Comment 2: I recommend the authors to give some background statement that motivated this work as the first sentence of the abstract.
Response 2: We added some background statement and rephrased the sentence.
“Microscale Ag/Fe bimetallic particles was prepared by depositing Ag on millimeter-scale 14 sponge iron particles and the reactivity for CCl4 removal from aqueous solution was tested.” was revised as “The cost-effective zero valent iron (ZVI) based bimetallic particle is a novel and promising technology for contaminant removal. The objective of this study was to evaluate the effectiveness of CCl4 removal from aqueous solution using microscale Ag/Fe bimetallic particles which was prepared by depositing Ag on millimeter-scale sponge ZVI particles.”
Comment 3: In the introduction section, the authors have missed citations on various critical sentences claiming many important things.
Response 3: Citations were added on the critical sentences.
Comment 4: Under the Materials and Methods section, some language issues with missing verbs in the statement. Please check well the word “were” just before purchased is missing.
Response 4: We rephrased section 2.1 as follows: “Chemicals (AgNO3, HCl, NaOH, CCl4, methanol, ethanol and sodium humic acid) used for this study were analytical reagent grade and obtained from the Sinopharm Chemical Reagent Company (Shanghai, China) and irregularly-shaped sponge iron particles with a size less than 150µm from Tianjin Zhongcheng iron powder factory (Tianjin, China) were used in the experiment.”
Comment 5: I recommend the authors to proof read again to iron out all these grammar and language issues as they exist throughout the manuscript.
Response 5: We corrected some errors in the manuscript.
Reviewer 2 Report
This is an important topic of research but the paper needs improvement. The major comment is that the authors do lots of fits (apparently) but there is no information about the equations used. It may be that many authors perform similarly, but this is not a good scientific practice regardless the research field.
Moreover, there are many parts of the paper that require better explanation or improvement.
Please find my comments in the annotated PDF.

Author Response
Dear Editors and Reviewers:
Thank you very much for your comments and suggestions. We have revised our manuscript marked with red after reading the comments provided by three reviewers. The main correction in the paper and responses to the reviewer’s comments as following.
Response to Reviewer #2:
Comment 1: Line 35 This is not clear for the broad audience: If it is considered to remove contaminants, how can it be toxic? Which toxic potential do you mean? Please specify.
Response 1: Many studies have demonstrated that the nZVI has toxic effects on the microbe. In Xie et al (2017) paper, Xie et al reviewed adverse effects of nZVI on microbial growth. The mechanisms for toxicity may include two aspects: one is physical damage: disruption of the cell membrane architectures, enhancement of membrane permeability; the other is biochemical destruction: interference in energy transduction and exchange, gene and protein damage. Therefore, “great toxic potential” was replaced with “potential ecotoxity”.
Xie Y, Dong H, Zeng G, et al. The interactions between nanoscale zero-valent and microbes in the subsurface environment: A review. 2017. Journal of Hazardous Materials, 321, 390-4017
Comment 2: Line 36 Here you should start a new paragraph. Moreover, better start the sentence with "Since recently, "
Response 2: We started a new paragraph with Since recently.
Comment 3: Line 46 You introduced the term "bimetallic system" for the first time here, but it is not clear to the reader why one should expect you to start explaining "bimetallic system" in this context. You should mention previously what is this and why it is important.
Response 3: “form bimetallic particles” was added in the previous sentence.
Comment 4: Line 55-56 Why should they have these values 1.36V and 1.24V? For which purpose? What do you mean with "should be" in this context? Please rephrase.
Response 4: The sentence was revised as “Theoretically, the potential difference for the Fe/Pd and Ag/Fe pairs are1.36V and 1.24V, respectively.”.
Comment 5: Line 83-91 Very confusing, difficult to read. Please add a table listening the conditions associated with each experimental set.
Response 5: We add a table to describe conditions of the experiment sets.
Four sets of batch experiments were conducted to investigate the reaction kinetics and effects of Ag/Fe dosage, solution pH and humic aicd on the dechlorination rates. The batch experiments were conducted in 100mL serum vials on a rotary shaker at 25±0.2℃ and 200±5r/min. The experiment conditions were listed in table 1.
Table 1. Experiment sets for CCl4 degradation by Ag/Fe particles
Experiment set |
pH |
CCl4 concentration (µg/L) |
Ag/Fe dosage (g/L) |
Ag loading on Ag/Fe (wt%) |
Humic acid concentration (mg/L) |
#1 |
7.0 |
2.0×104 |
20 |
0.2, 0.4, 0.6, 0.8, 1.0 |
0 |
#2 |
7.0 |
2.0×104 |
5, 10, 20, 30, 40 |
0.4 |
0 |
#3 |
4.0, 5.0, 6.0, 7.0, 8.0 |
2.0×104 |
20 |
0.4 |
0 |
#4 |
7.0 |
2.0×104 |
20 |
0.4 |
0, 5, 10, 20 |
Comment 6: Line 131-132 What do you mean with "In general"? Do you mean that other authors found this result? Or is this the result of Fig. 3? If the latter is the case, then you should write: "As one can see from this figure, (...)".
Response 6: It means the results of Fig.3. “In general” was revised as “As can be seen from this figure”
Comment 7: Line 135-137 If this was found before, then it would be helpful to the reader if you could add a short sentence explaining why this happens.
Response 7: It was found before. A sentence “At higher Ag loadings, the removal efficiency decreased due to larger aggregation of Ag, which reduced the Ag surface area.” was added.
Comment 8: Line 145-146 Again this is not clear what is your result and what has been found before. Please tell explicitly to the reader what is new in your results compared to Ref. 22.
Response 8: It was found before. A sentence “For example, the percent removal of CCl4 decreased from 83.6% to 71.9% when the Ag loading increased from 0.4wt% to 1.0wt% at 20min.” was added to describe the our results.
Comment 9: Line 228 What are the lines? Are they fits? If yes, then please provide the equations of these fits as well as the values of all parameters in the paper.
Response 9: The lines were fitted. The equations were added in table 3. The Figure 4 in line 197 did the same.
Comment 10: Line 278 What are these lines? Are they fits? What are the equations? What are the values of the fit parameters you obtain from the best fit?
Response 10: The lines were fitted. The lines were deleted.
Comment 11: Line 304 What are the functions that you use to fit the measurements in each plot? Please provide the equations.
Response 11: Three sentences were added in Section 3.1 to describe how to fit the XPS data. “The test results were processed with XPSPEAK4.1 and a Shirley function was used to subtract the background. The peaks of O1s and Fe2p were fitted with Lorentzian Gaussian curves. The binding energy scale was corrected using the C1s signal of 285.19 eV.”
Comment 12: Line 323 What are the lines? Are the fits? What are the equations of these fits? What are the values of the fit parameters?
Response 12: The lines were fitted. The lines were deleted.
Comment 13: Line 336 Please add a paragraph telling what are the main novelties compared to previous publications related to this topic (3-4 points would be enough). Why is this paper important and should be published? Note that you wrote a motivation in the introduction. Therefore, how is this paper contributing to solve the problems, open issues and challenges raised in the intro?
Response 13: One paragraph was added. “The iron-based microscale particle with a transition metal developed in this study possesses ability to substantially accelerate the dechlorination reaction and can be used to degrade organic contaminants such as CCl4. Additionally, it has long life span, low ecotoxicity and can be prepared economically.”

Reviewer 3 Report
These are comments. The manuscript is overall excellent. It would make a nice assignment to a graduate level, especially when it comes to the XPS results.
Line 17 What is ZVI? Assume sponge. Defined on line 28. Seems like should be defined in the abstract if reading abstract entices reader to read whole paper.
Line 31 Might use smaller, as better word.
Line 54 Since rust is mostly Fe (III) oxide and Fe (III) hydroxide, I thought, why do you give the reduction potential for Fe (II)? I am wrong, we start at zero valence Fe. Rust is an unwanted item.
Line 79 In reduction at surface, why does the reduced metal stay? I never thought about it.
Line 93 A little unclear. 100µL samples extracted by 100µL syringes? Confusing, might consider “One hundred µL samples were placed… But fine as is.
Line 130 shows
Line 133 the paper referred to used nZVI?
Line 201 Interesting detecting of chloroform as intermediate.
Line 234 Nice demo of 2 stage chemical reaction.
Line 262 pH demonstration also nice
Line 312 Has often been called most commonly studied substances in the world.
Author Response
Thank you very much for your comments and suggestions. We have revised our manuscript marked with red after reading your comments.
Comment 1: Line 17 What is ZVI? Assume sponge. Defined on line 28. Seems like should be defined in the abstract if reading abstract entices reader to read whole paper.
Response 1: We defined the ZVI in the first sentence of the abstract. “The cost-effective zero valent iron (ZVI) based bimetallic particle is a novel and promising technology for contaminant removal.”
Comment 2: Line 31 Might use smaller, as better word.
Response 2: As suggested, the “small” was replaced with “smaller”.
Comment 3: Line 79 In reduction at surface, why does the reduced metal stay? I never thought about it.
Response 3: The reduced metal was deposited on the ZVI surface.
Comment 4: Line 93 A little unclear. 100µL samples extracted by 100µL syringes? Confusing, might consider “One hundred µL samples were placed… But fine as is.
Response 4: The sentence was revised as: The samples (100µL) were taken from the serum vials and then placed in 20mL headspace vials with 9.9 mL deionized water at selected time intervals.
Comment 5: Line 133 the paper referred to used nZVI?
Response 5: The ZVI in this study was microscale ZVI. We added a word of “microscale” before the “sponge ZVI”
Round 2
Reviewer 2 Report
The authors did a great job in revising their article following the reviewers' comments and the article can be accepted in present form.